# Barley Cultivar Sarab 1 Has a Characteristic Region on the Thylakoid Membrane That Protects Photosystem I under Iron-Deficient Conditions

**DOI:** 10.3390/plants12112111

**Published:** 2023-05-26

**Authors:** Akihiro Saito, Kimika Hoshi, Yuna Wakabayashi, Takumi Togashi, Tomoki Shigematsu, Maya Katori, Takuji Ohyama, Kyoko Higuchi

**Affiliations:** Laboratory of Biochemistry in Plant Productivity, Department of Agricultural Chemistry, Tokyo University of Agriculture, Setagaya-ku, Tokyo 156-8502, Japan; a3saito@nodai.ac.jp (A.S.); to206474@nodai.ac.jp (T.O.)

**Keywords:** barley, iron deficiency, LHCII, photosystem I, thylakoid membrane

## Abstract

The barley cultivar Sarab 1 (SRB1) can continue photosynthesis despite its low Fe acquisition potential via roots and dramatically reduced amounts of photosystem I (PSI) reaction-center proteins under Fe-deficient conditions. We compared the characteristics of photosynthetic electron transfer (ET), thylakoid ultrastructure, and Fe and protein distribution on thylakoid membranes among barley cultivars. The Fe-deficient SRB1 had a large proportion of functional PSI proteins by avoiding P700 over-reduction. An analysis of the thylakoid ultrastructure clarified that SRB1 had a larger proportion of non-appressed thylakoid membranes than those in another Fe-tolerant cultivar, Ehimehadaka-1 (EHM1). Separating thylakoids by differential centrifugation further revealed that the Fe-deficient SRB1 had increased amounts of low/light-density thylakoids with increased Fe and light-harvesting complex II (LHCII) than did EHM1. LHCII with uncommon localization probably prevents excessive ET from PSII leading to elevated NPQ and lower PSI photodamage in SRB1 than in EHM1, as supported by increased Y(NPQ) and Y(ND) in the Fe-deficient SRB1. Unlike this strategy, EHM1 may preferentially supply Fe cofactors to PSI, thereby exploiting more surplus reaction center proteins than SRB1 under Fe-deficient conditions. In summary, SRB1 and EHM1 support PSI through different mechanisms during Fe deficiency, suggesting that barley species have multiple strategies for acclimating photosynthetic apparatus to Fe deficiency.

## 1. Introduction

A large amount of iron (Fe) is required for efficient electron transfer in the photosynthetic apparatus. Specifically, the photosystem I (PSI) reaction center harbors three 4Fe-4S clusters and is the primary target of Fe deficiency [1,2,3,4,5]. Two strategies to maintain photosynthesis in Fe-deficient chloroplasts are possible: a continuous supply of Fe to reaction center proteins or the reorganization of protein complexes to ensure electron transfer utilizing a smaller amount of Fe. Barley cultivar Sarab1 (SRB1) has excellent tolerance to Fe deficiency among more than 20 barley cultivars originating worldwide based on Photosynthetic Iron-Use efficiency (PIUE) developed recently in our study on the chloroplast Fe economy [6]. Using quantification of the Fe uptake rate using the live autography system, unexpectedly, SRB1 exhibited a lower rate of Fe acquisition into developing leaves with a smaller accumulation of reaction center proteins of both PSI and photosystem II (PSII) under the Fe-deficient condition than other cultivars [7]. In contrast, another Fe deficiency-tolerant cultivar Ehimehadaka 1 (EHM1) accumulated more Fe in shoots and maintained more amounts of reaction center proteins than did SRB1 under Fe-deficient conditions [7]. However, the expression of some of the genes involved in the Fe-S cluster supply pathway to photosystems, including that of sulfur utilization factor (SUF), reduced similarly under Fe deficiency in both SRB1 and EHM1 [7]. These results revealed that active Fe acquisition into shoots or regulation of the Fe-S delivery system under Fe-deficient conditions is not always the main factor contributing to Fe deficiency tolerance [7].

Several important findings have been reported regarding the adaptation of the plant photosynthetic apparatus to Fe deficiency, mainly using algae as research material. In cyanobacteria, the “iron-stress-induced” gene *isiA* is expressed, and the product IsiA protein (CP43′) forms a giant 18-subunit ring around the trimeric PSI core complex under Fe deficiency [8]. Recently, an analysis of energy transfer in the PSI-IsiA supercomplex indicated that IsiA functions as an energy donor but not as an energy quencher in the supercomplex [9]. In addition to PSI, to protect the acceptor side of PSII against Fe deficiency in cyanobacteria, an additional pigment–protein complex, IdiA (iron deficiency-induced protein), is expressed [10]. In the obligate photoautotrophic alga *Dunaliella salina*, Fe deprivation induces the expression of a chlorophyll *a*/*b*-binding protein Tidi, similar to that of IsiA protein. Tidi resembles the light-harvesting antenna complex protein of PSI (LHCI) and acts as an accessory antenna of PSI [11]. Additionally, in another eukaryote algae *Chlamydomonas reinhardtii*, Fe deprivation causes the remodeling of LHCI and decreases the antenna size of PSI to reduce the efficiency of excitation energy transfer between LHCI and PSI [12,13]. The stress-inducible light-harvesting antenna LHCSR3 protein was also expressed under conditions of Fe deficiency in *C. reinhardtii*, leading to increased nonphotochemical quenching (NPQ), thereby providing protection from photoinhibition [14]. Although NPQ is most often related to PSII photoprotection, it also protects PSI through quenching of the LHCII antenna pool functionally associated with PSI. Along with the regulation of these antenna systems, the chloroplasts in *Chlamydomonas* change the Fe economy to preferentially maintain the Fe-containing enzyme Fe superoxide dismutase by balancing the rates of synthesis and degradation of many plastid Fe-proteins [15,16].

In contrast to algae, higher plants lack the Fe deficiency-induced light-harvesting antennae proteins such as IsiA, Tidi, or LHCSR3 in their genomes, making it challenging to develop a link between photosynthesis and Fe deficiency. Nevertheless, some plant species, including barley, can maintain photosynthetic function even after prolonged exposure to Fe deficiency [17]. In this context, we have shed light on the diversity of Fe-deficient responses on photosystems in Graminaceae plants and revealed the barley-specific photoprotective mechanism using light-harvesting antenna Lhcb1 isoforms during Fe deficiency [5,17,18]. Interestingly, we also found that SRB1, a barley variety with significantly higher Fe deficiency tolerance, may have a unique electron transfer function to protect downstream PSI [6]. However, the details surrounding this mechanism still need to be discovered.

In this work, we investigated the characteristics of photosynthetic electron transfer and Fe and protein distribution on thylakoid membranes in the barley cultivar SRB1. Thereafter, we compared these characteristics between SRB1 and other barley cultivars under Fe-deficient conditions. Among the barley cultivars we used, EHM1 was selected as the reference cultivar for Fe deficiency tolerance. This specific cultivar was selected to investigate the SRB1-specific tolerant mechanism and obtain physiologically meaningful data under severe Fe-limited conditions as other cultivars are Fe deficiency-susceptible and cannot maintain photosynthesis under such severe Fe-deficient conditions. Through comparative analysis, we explore the characteristics of SRB1 that contribute to the maintenance of photosynthesis under limited Fe.

## 2. Results

### 2.1. SRB1 Has Superior Ability to Suppress Electron Transfer Downstream of PSI

SRB1 maintained the photosynthetic electron transfer function downstream of PSII, including cytochrome (cyt) *b_6_f*, and PSI, through an unknown mechanism despite having low Fe and very small amounts of reaction-center proteins in its leaves under Fe-deficient conditions [6,7]. To elucidate this, we analyzed PSII and PSI simultaneously using Dual-PAM-100 among four cultivars (SRB1 and EHM1 for tolerant cultivars, ETH2 and MSS for susceptible cultivars) with different Fe deficiency tolerance levels, as identified in a previous study [6].

The leaves of Fe-deficient and susceptible barley cultivars often exhibit highly severe chlorotic and wilt symptoms. Further, we cannot observe physiologically meaningful chlorophyll fluorescence under the same Fe-deficient condition as Fe deficiency-tolerant cultivars. We used the same cultivation techniques applied in previous reports [6,7]. Appendix A shows a typical cultivation result: chlorophyll content in all four cultivars under Fe-sufficient conditions was 1.5 mg/gFW. The chlorophyll content under Fe-deficient conditions was 0.3–0.5 mg/gFW in all four cultivars with no severe necrosis spots (Appendix A). Fe content in leaves was also comparable among all cultivars under Fe-deficient conditions (Appendix A); thus, we successfully prepared plant materials exhibiting almost the same extent of Fe deficiency chlorosis and Fe content as previously reported [6].

The maximum quantum yield of PSII, denoted by Fv/Fm, was around 0.8 for all four cultivars under Fe-sufficient conditions (Figure 1A), equivalent to the theoretical value required for healthy leaves. EHM1 did not reduce Fv/Fm due to Fe deficiency among four cultivars. In contrast, Fv/Fm in the Fe-deficient leaves compared was slightly, but significantly, decreased compared to that in the Fe-sufficient leaves in SRB1, ETH2, and MSS varieties (Figure 1A). The quantum yield of regulated energy dissipation, Y(NPQ), increased with Fe deficiency compared with Fe-sufficient conditions in all cultivars (Figure 1B). This is consistent with the results of our previous report that Fe-deficient barley induces NPQ to dissipate excess light energy as heat to avoid PSII photoinhibition resulting from Fe-deficiency-mediated defects in electron transport, regardless of the barley variety [6,17]. Interestingly, SRB1 had higher Y(NPQ) than the other cultivars under both Fe-sufficient and Fe-deficient conditions. The NPQ of MSS, the most susceptible cultivar [6], was also similar to that of SRB1 under Fe-deficient conditions (Figure 1B). The quantum yields of non-regulated energy dissipation related to PSII photoinhibition, Y(NO), were not significantly different among the cultivars, irrespective of the Fe nutritional status (Figure 1C), and Fe deficiency increased the absolute Y(NO) in all cultivars. Therefore, the degree of PSII photoinhibition in all varieties under Fe-deficient conditions was comparable. Thus, consistent with the results of our previous report [6], the photoprotective mechanism of PSII through NPQ induction in Fe-deficient leaves can be considered a ubiquitous Fe-acclimation system for barley species, confirming that PSII maintenance itself is not a main factor in imparting differential Fe deficiency tolerance within barley cultivars.

Next, to clarify the status of PSI, P700 absorbance was examined; P700 maximum oxidation capacity (Pm) is generally used as an indicator of PSI quantity and function because it reflects the maximum absorbance of P700. Pm values in all cultivars were dramatically reduced under Fe-deficient conditions compared to those under Fe-sufficient conditions (Figure 1D), confirming previous findings that Fe deficiency primarily affects PSI [2]. The Fe deficiency tolerant cultivars SRB1 and EHM1 had comparable Pm values under Fe-deficient conditions and significantly higher Pm values than those of the Fe deficiency-susceptible varieties ETH2 and MSS, suggesting that functional PSI maintenance is an essential factor in Fe deficiency tolerance.

Y(ND) is a PSI donor-side limitation, a mechanism of PSI protection that inhibits electron transfer from PSII to PSI by NPQ, plastoquinone (PQ) reduction, or ΔpH expansion in the thylakoid lumen. SRB1 showed the highest Y(ND) under Fe deficiency among the four cultivars, followed by MSS with equally high Y(ND) (Figure 1E). Y(NA) is a parameter for PSI acceptor-side limitation related to PSI photoinhibition by P700 over-reduction and subsequent generation of the toxic reactive oxygen species (ROS). Interestingly, Y(NA) increased under Fe deficiency in all cultivars except SRB1. In contrast, SRB1 showed almost no increase in Y(NA), even under Fe-deficient conditions, indicating that SRB1 suppressed P700 over-reduction.

These results suggest that the mechanism of the P700 oxidation system [19] is the reason for the ability of SRB1 to maintain photosynthesis during Fe deficiency. There is a link between elevated NPQ and increased Y(ND) [20]. This linkage is due to decreased linear electron transfer from PSII to PSI by increased thermal dissipation of light energy. Consistent with this, Y(NPQ) was highest in SRB1 under Fe deficiency (Figure 1B), and Y(ND) was concomitantly elevated under Fe deficiency (Figure 1E), suggesting that the high P700 oxidation induction of SRB1 is related to NPQ induction. To further confirm this observation, the related parameters were also taken into consideration. 1-qL, which indicates the reduced state of PQ pools, was higher in SRB1 than in the other cultivars (Appendix A). Further, the Y(I)/Y(II) ratio, the ratio of the quantum yield of PSI to PSII, was significantly higher in SRB1 under Fe-deficient conditions (Appendix A), indicating that the PSII electron transfer rate was kept low relative to the PSI electron transfer rate. Therefore, a safe PSII–PSI excitation balance that is less prone to PSI photoinhibition can be maintained in the Fe-deficient SRB1. Although MSS also had high Y(NPQ) and Y(ND) like SRB1 under Fe-deficient conditions (Figure 1E), the absolute Y(NA) of Fe-deficient MSS was twice that of Fe-deficient SRB1 (Figure 1F), indicating that PSI photoinhibition is unavoidable in the Fe-deficient leaves of MSS. Interestingly, the Fe-sufficient leaves of MSS exhibited higher Y(NA) than those of other cultivars (Figure 1F; Fe-sufficient SRB = 0.63, Fe-sufficient MSS = 0.136), suggesting that even before Fe deficiency, MSS was not highly effective in PSI photoprotection via the P700 oxidation system, presumably because of the low PSI structural stability associated with low Fe availability in the chloroplasts, as previously reported [6].

### 2.2. SRB1 Is Excellent at Maintaining Functional PSI under Fe-Deficient Conditions

To determine whether the Fe-deficient cultivar SRB1 maintains a higher accumulation of PSI proteins than the other cultivars, we quantified the contents of major photosystem proteins of both PSII and PSI. The amounts of PSII core subunit proteins D1 and D2 were generally similar among cultivars under Fe-sufficient conditions (Figure 2A,B). However, because the Western blot (Figure 2A) was based on data obtained from different electrophoresis gels for each variety, the proteins extracted from each variety grown under Fe-sufficient conditions were also electrophoresed on the same gel, confirming that the PSII amount was equivalent among the four cultivars. On the other hand, the D1 and D2 protein amounts in Fe-deficient SRB1 leaves were significantly lower (approximately 30%) compared with those in Fe-deficient EHM1 or ETH2 leaves. The other PSII core subunit, cyt *b_559_* (PsbE), which contains a hem Fe, did not differ significantly among cultivars regardless of Fe sufficiency or Fe deficiency. Thus, although SRB1 contained lower amounts of D1 and D2 than those in other cultivars, the electron transfer function on PsbE within the PSII complex was maintained under Fe-deficient conditions, consistent with the fact that *F*v/*F*m was not greatly reduced by Fe deficiency (Figure 1A). These results substantiate that PSII is not the primary reason for the difference in Fe deficiency tolerance levels among these cultivars.

In the case of PSI core subunits, the amounts of PsaA, PsaB, and [4Fe-4S]-containing PsaC in the Fe-deficient SRB1 were unexpectedly and significantly decreased by about 10% compared to their levels in Fe-sufficient control (Figure 2A,C). This residual percentage of PSI core subunits in SRB1 was equal to or less than that of the other Fe deficiency-susceptible cultivars ETH2 and MSS, in which considerably fewer PSI reaction center proteins could be detected. In contrast, EHM1, another Fe deficiency-tolerant variety, retained about 30% of PsaA and as much as 20–25% for PsaB and PsaC under Fe-deficient conditions. As the Western blot analysis in Figure 2A was performed after adjusting the protein concentration to be the same in both Fe-sufficient and Fe-deficient samples, only a tiny amount of PSI protein could be detected in the Fe-deficient samples due to chlorosis. Because of concerns about the quantitation limit of Western blot analysis on protein content basis in Figure 2, we re-performed the same analysis on samples in which the chlorophyll concentration was adjusted to make them equal among all samples to better detect thylakoid membrane proteins in the Fe-deficient samples (Appendix A). As shown in Appendix A (Western blot on a chlorophyll content basis), the reduction in PSII and PSI core subunits was significantly observed in SRB1 and the Fe deficiency-susceptible cultivars ETH2 and MSS. In contrast, the decrease in both PSII and PSI was less pronounced in EHM1, supporting the results shown in Figure 2. These results shown in Appendix A (Western blot on a chlorophyll content basis) reconfirmed the data shown in Figure 2 (Western blot on a protein content basis); unlike SRB1, it is evident that EHM1 has a strategy to maintain the total amount of photosystem proteins during Fe deficiency.

Because SRB1 has only a small amount of PSI reaction centers (Figure 2A,C and Appendix A), we speculate that the remaining PSI in the Fe-deficient SRB1 may be more functional than those in the other three cultivars. PSI functionality has generally been calculated based on the ratio of Pm before and after PSI photoinhibition treatments [21]. In relation to this concept, we attempted to calculate PSI functionality by using the ratio of Pm under the Fe-deficient conditions to the Pm of leaves under the control (Fe-sufficient) conditions. The problem is that in the case of Fe deficiency, Pm is considerably lower, influenced by a decrease in PSI complexes, regardless of PSI functionality. To eliminate the influence of the decline in PSI content in Fe-deficient leaves, we divided the Pm value by the relative accumulation of PSI core proteins, PsaA, PsaB, and PsaC, respectively (Figure 2C), for normalization to align PSI content computationally under Fe-sufficient and Fe-deficient conditions. As a result of this modified method from Lempiäinen et al. [21], we have successfully calculated PSI functionality under Fe-deficient conditions (Figure 2D). As shown in Figure 2D, the residual functional PSI rate of the Fe-deficient SRB1 was approximately two times higher than that of the other three cultivars, EHM1, ETH2, and MSS. This result was also confirmed in Appendix A (Western blot on a chlorophyll content basis), i.e., the functional PSI rate of SRB1 under Fe-deficient conditions was about two-fold higher than in other varieties.

These results suggest the different strategies between the two Fe deficiency-tolerant cultivars: SRB1, which keeps a large proportion of functional PSI to overcome the decrease in PSI accumulation, and EHM1, which maintains a sufficient amount of PSI proteins to preferentially bind Fe-S clusters. Unlike the tolerant cultivars, EHM1 and SRB1, the susceptible cultivars, ETH2 and MSS, exhibited low PSI accumulation (Appendix A) or the functional PSI ratio (Figure 2D and Appendix A). The low PSI stability under Fe-deficient conditions in these varieties may be related to various systems and would not be necessarily related to photosystem function. Therefore, we excluded these cultivars from further biochemical analyses focusing on the thylakoid membranes.

### 2.3. Organization of Thylakoid Membrane in Fe-Sufficient Leaves

With regard to the differences in the amount of functional PSI maintained under Fe-deficient conditions among varieties, we investigated the sequence of the *PsaC* gene, whose product harbors two 4Fe-4S clusters in PSI, based on the query whether the differences in the primary structure itself may be responsible for PSI’s Fe availability. However, the gene encoding PsaC is a single-copy gene in the barley genome and no sequence differences were detected between SRB1 and EHM1 using Sanger dideoxy sequencing. Thus, differences in its structure or ability to bind the Fe cofactors are not likely to explain the different responses in photosystems to Fe deficiency between SRB1 and EHM1.

Therefore, to focus on whether thylakoid membrane structure and its protein distribution affect Fe availability, the analysis using transmission electron microscopy (TEM) was conducted to observe the organization of grana stack and stromal thylakoids in chloroplasts of SRB1 and EHM1. Although the structures of thylakoids in Fe-deficient mesophyll cells varied from rather normal to swollen abnormal thylakoids [5,18], we found that SRB1 seemed to have more stromal lamellar sheets than did EHM1, even under the Fe-sufficient condition. To confirm this characteristic quantitatively, we tried to simply estimate the stromal thylakoid/grana stacks on the TEM images.

In this analysis, the central part of the newest fully expanded leaf was used for imaging analysis as the same position used in Figure 1 and Figure 2. It is known that there are two types of stroma lamellae membranes: one is directly linked to the grana stacks, and the other is like a large sheet that is not directly linked to the grana structure [22]. The former type, called “stroma lamella-grana (SG) structure”, is responsible for the compartmentalization of PSI and the restoration of PSII [23]. Therefore, only stromal lamellae in the SG structure were selected for this analysis as structures that have a significant effect on photosynthesis.

The granal and stromal thylakoid membrane lengths were traced on the vertically visual image of thylakoids using NIH ImageJ (Figure 3 and Appendix A). More than twenty representative SG structures (*n* = 21 for EHM1 and *n* = 24 for SRB1) in four TEM images for each cultivar were analyzed (Appendix A). As shown by the yellow line in Figure 3A, the stroma lamella (the non-appressed region) includes the top and bottom planes of the grana stacks and the sheet structure connected with the grana stack (Figure 3A). After calculating the stromal thylakoid to granal thylakoid ratio, we confirmed that the ratio for EHM1 was about 0.6, while that for SRB1 was about 0.8, which in turn showed that the ratio of SRB1 was significantly higher than that of EHM1 by about 45% (Figure 3B).

### 2.4. Protein Complexes Contained in Thylakoid Fractions with Different Densities in Fe-Sufficient Leaves

As shown above, we observed differences in the organization of the thylakoid membrane between the two cultivars using TEM. We also found experimentally that the supernatant obtained during the thylakoid membrane extraction procedure from SRB1 leaves always showed more chlorophyll than that from EHM1. The thylakoid membranes in the supernatant of SRB1 were probably derived from unstacked thylakoid membranes that were physically disrupted during homogenization and could not be precipitated after the normal low-speed centrifugation. Therefore, we analyzed the composition of such thylakoid membranes. We designated the slurries obtained after the 2500× *g* centrifugation as “heavy/high-density thylakoids (H-Thy),” corresponding to typical thylakoid membranes usually analyzed. In contrast, the remaining thylakoids in the supernatant were recovered as pellets after centrifugation at 10,000× *g*, designated as “light/low-density thylakoids (L-Thy).”

After obtaining H-Thy and L-Thy from Fe-sufficient leaves of SRB1 and EHM1, we applied each fraction to sucrose density gradient (SDG) centrifugation following solubilization with n-dodecyl β-D-maltoside (β-DM). The appearance of SDG tubes and the composition of proteins of H-Thy were highly similar between SRB1 and EHM1 (Figure 4). This experiment was also performed on thylakoids isolated from stored frozen leaves, and the SDG analyses were generally reproducible (Appendix A). Total proteins and Fe distribution patterns, not absolute values, were almost identical among the two cultivars. In this analysis, we excluded the eighth fraction (fr.8) from consideration of Fe since the lower fractions were less reproducible for quantitative Fe. For example, fr.8 of SRB1 had a large proportion of Fe, as shown in Figure 4, but the corresponding fr.8 obtained from snap-frozen leaves did not have such an amount of Fe (Appendix A). Such irregular Fe detected from fr.8 of SRB1 in Figure 4 may not have biological meanings. Note that PSII fractions in H-Thy had more Fe than did PSI-LHCI fractions in both SRB1 and EHM1 (Figure 4), but this is because this collected fraction contains both PSII and PSI as confirmed by Western blot analysis (Figure 4 and Appendix A) and is not considered abnormal data.

The appearance and protein distribution pattern of the SDG tubes of L-Thy in SRB1 matched those of H-Thy. However, the Fe contents of fr.3 (LHCII-like fraction) and fr.7 (PSI-LHCI-like fraction) in L-Thy of SRB1 were higher than those in the corresponding fractions in H-Thy of SRB1 (Figure 4), suggesting that more Fe seemed to be allocated to the L-Thy in SRB1. In contrast, in EHM1, the distribution of green bands in L-Thy and H-Thy was different in appearance, with fewer fr.7 bands corresponding to PSI-LHCI in L-Thy than in H-Thy. Indeed, the signals of the Western blot against anti-PsaA, B, and C in fr.7 of L-Thy of EHM1 were weak when compared to the corresponding fraction (PSI-LHCI) of H-Thy of EHM1 (Figure 4).

Besides that, three differences between SRB1 and EHM1 were found in both H-Thy and L-Thy. First, the distributions of PsaA, B, and C extended to fr.8 in the case of SRB1, whereas it extended to the upper layer where fr.5 is located in the case of EHM1 (Figure 4). Thus, the thylakoid membrane of SRB1 may contain PSI complexes of larger molecular weight than those of EHM1, regardless of H-Thy and L-Thy. Second, signals of anti-Lhcb1 in both H-Thy and L-Thy from SRB1 thylakoid membranes were enlarged when compared to those from EHM1, but signals of anti-Lhcb2 were not (Figure 4). Since total protein amounts in the LHCII fractions in H-Thy and L-Thy of EHM1 were not smaller than those of SRB1 (Figure 4) and the intensities of CBB stain of LHCII proteins did not differ between the two cultivars (Appendix A), the abundance of Lhcb1 in LHCII should be higher in SRB1 than in EHM1. Third, LHCII fractions from H-Thy of SRB1 and the corresponding fr.3 from L-Thy of SRB1 had larger proportions of Fe than did adjacent fractions when compared to those of EHM1.

It is difficult to recover large quantities of thylakoid membranes from the chlorotic Fe-deficient fresh leaves due to the size scale of the experimental apparatus. Therefore, the same experiment was conducted in parallel on stocked frozen leaves to obtain the reproducibility of the data. Data of H-Thy from frozen leaves (Appendix A) were almost the same as that from fresh leaves (Figure 4). Minor differences between fresh and frozen leaves were found; a few amounts of the extra green bands appeared in fr.8 of H-Thy from EHM1 (Appendix A); also, fr.7 of L-Thy of frozen SRB1, which may correspond to PSI-LHCI, was decreased more than that of H-Thy of fresh SRB1 when comparing the appearance of the green band and the amount of PsaA, B, and C (Figure 4 and Appendix A). Meanwhile, the following differences were well reproducible in frozen and fresh leaves: the differences in the distribution patterns of PsaA, B, and C and signal intensities of anti-Lhcb1 (Figure 4) were almost reproducible in the case of frozen leaves (Appendix A); the LHCII fraction of SRB1 H-Thy and fr.3 of SRB1 L-Thy had a relatively higher proportion of Fe than that of EHM1 (Appendix A), similar to the data of fresh SRB1 leaves (Figure 4); the relative composition of Lhcb1, which was observed to be higher in SRB1 than in EHM1, was also reproducible among frozen leaves (Appendix A).

### 2.5. The Influences of Fe Deficiency on Protein Distribution in the Thylakoid Membrane

The SDG centrifugation method, which requires large amounts of thylakoid, is disadvantageous for further analysis because the recovery of the L-Thy fraction is further reduced in the case of Fe-deficient leaves. Therefore, we tried to use Native-PAGE, which generally requires fewer thylakoid proteins. However, Fe contamination from the PAGE gel prevented accurate analysis of trace amounts of Fe. Instead, we quantitatively investigated the influences of Fe deficiency on the partitions of chlorophyll, Fe, and proteins between H-Thy and L-Thy.

We used twice the amounts of leaves to isolate the thylakoid membrane from chlorotic leaves compared to those from control leaves since the accumulation of protein complexes on the thylakoid membrane was remarkably decreased by Fe deficiency, even though both SRB1 and EHM1 are Fe deficiency-tolerant cultivars. The recovery rate of chlorophyll is shown in Appendix A. Fe deficiency clearly decreased the amount of H-Thy but not L-Thy in both cultivars based on chlorophyll (Figure 5A,B); that is, Fe deficiency increased the ratio of L-Thy to H-Thy. Fe deficiency may result in the loosening and easy disintegration of the structure of thylakoid membranes by decreasing the amount of protein complexes on them. In fact, we reported abnormal thylakoid membranes in Fe-deficient leaves of EHM1 (moderate chlorosis: [18], severe chlorosis: [5]). However, the ratio of L-Thy to H-Thy differed between the two cultivars. SRB1 exhibited a higher ratio (0.10; SE 0.007) of L-Thy to H-Thy than that of EHM1 (0.05; SE 0.015), even under the control condition based on chlorophyll when values of H-Thy were 1 (Figure 5A,B, green and green-hatched bars). This difference was increased by Fe deficiency, up to 0.46 (SE 0.12) for SRB1 and up to 0.21 (SE 0.031) for EHM1 (Figure 5A,B, orange and orange hatched bars). Based on the amounts of chlorophyll, it was estimated that one-third of the thylakoid membrane derived from Fe-deficient SRB1 leaves was L-Thy. We also determined the Fe content of each fraction. The ratio of Fe in L-Thy to Fe in H-Thy was 0.11 (SE 0.005) and 0.11 (SE 0.018) for SRB1 and EHM1, respectively, when grown under control conditions; and 0.34 (SE 0.25) and 0.20 (SE 0.034) for SRB1 and EHM1, respectively (Figure 5C,D), when grown under Fe-deficient conditions. These results indicate that SRB1 had a relatively high chlorophyll/Fe ratio in L-Thy than in H-Thy under Fe-deficient conditions.

Furthermore, we evaluated the amounts of proteins on the thylakoid membranes. Since Fe deficiency drastically decreases protein complexes on the thylakoid membrane (Figure 2A), we applied four-fold amounts of Fe-deficient samples per lane for Western blot analysis to observe differences between H-Thy and L-Thy. CBB stain images are shown in Appendix A. The signal intensity of the Western blot was evaluated using ImageJ and we calculated the ratio of L-Thy to H-Thy. We did not normalize the values obtained from Fe-deficient samples based on those from control samples because we had to load eight-fold amounts of Fe-deficient materials to obtain enough signals to compare with those from the control materials, as described above. The Fe-containing proteins, D1 and D2 of the PSII reaction center and PsaA, PsaB, and PsaC of the PSI reaction center, were detected using specific antibodies. Anti-HvLhcb1 and Lhcb2 antibodies were also used, as we previously reported the migration of HvLhcb1 under Fe-deficient conditions [18]. Allocations of all proteins detected in L-Thy were increased by Fe deficiency in both cultivars, but the proportion of each protein on H-Thy and L-Thy was largely different between the two cultivars (Figure 5E). A large proportion of all detected proteins was localized on H-Thy in EHM1 (Figure 5E and Appendix A). In contrast, SRB1 allocated more D1, D2, Lhcb1, and Lhcb2 belonging to PSII to L-Thy than did EHM1, regardless of the Fe nutritional status (Figure 5E and Appendix A). However, PsaA, PsaB, and PsaC of PSI were mainly localized on H-Thy of the control SRB1, similar to that of EHM1, but were dramatically decreased by Fe deficiency (Figure 5E and Appendix A), consistent with Figure 1D. The proportions of these PSI core proteins on L-Thy, however, were further increased by Fe deficiency in SRB1 than in EHM1. Typically, PSII has relatively lower Fe content than PSI. Thus, unbalanced distributions of reaction center proteins composing PSII, PSI, and LHCII proteins on L-Thy obtained from SRB1 (Figure 5E) were consistent with a higher chlorophyll/Fe ratio in L-Thy than in H-Thy extracted from Fe-deficient SRB1 (Figure 5A,C).

We tested the protein distribution on the thylakoid membrane of MSS since we observed a significant reduction in PSI core proteins in both Fe deficiency-tolerant SRB1 and Fe deficiency-susceptible MSS (Figure 2A). The proportions of H- and L-Thy in control and Fe-deficient leaves of MSS were similar to those of SRB1 (Appendix A). PSII core proteins (D1 and D2) and PSI core proteins (PsaA, PsaB, and PsaC) were more distributed to L-Thy than that in EHM1, similar to SRB1 (Appendix A and Figure 5E). The distribution of LHCII proteins to H- and L-Thy in MSS exhibited an intermediate pattern between SRB1 and EHM1 (Appendix A and Figure 5E).

## 3. Discussion

### 3.1. Electron Transfer around PSI under Fe-Deficient Conditions Is Better in Sarab1 than in Other Cultivars

We previously reported that the ability of Fe deficiency-tolerant cultivars of barley to increase PIUE is related to the optimization of the electron flow downstream of PSII, including cyt *b*_6_*f* and PSI [6]. Consistent with this report, we found that the two Fe deficiency-tolerant cultivars, SRB1 and EHM1, both maintained higher Pm values (Figure 1D) than did the Fe deficiency-sensitive variety under Fe-deficient conditions. However, the mechanism of maintaining PSI function appeared to differ between the two cultivars, with SRB1 adopting a strategy dependent on the maintenance of functionality rather than the accumulation of PSI proteins, whereas EHM1 maintains more PSI protein complexes with lower functionality than SRB1.

The strategy of SRB1 for maintaining functional PSI is a P700 oxidation-inducing system [19,24] that increases P700^+^ (Figure 1E,F). Two central mechanisms are known to be involved in the P700 oxidation induction. The first is suppression of the accumulation of reduced P700* by decreasing the electrons transferred to P700 by suppressing the PSI donor [24,25]. This mechanism includes NPQ induction in PSII, ΔpH in the thylakoid membrane lumen, and functional inhibition of the PQ pool [20]. In the present study, 1-qL, which indicates the reduced state of the PQ pool, was relatively higher in SRB1 than in other cultivars, both in Fe-sufficient and Fe-deficient plants (Appendix A), suggesting that the PQ pool in SRB1 was likely reduced by an increase in the ΔpH prior to exposure to Fe deficiency. Such highly reduced states of the PQ pool could activate the xanthophyll cycle to increase zeaxanthin, an efficient heat-dissipating pigment [26] and STN7 kinase to enhance LHCII protein phosphorylation [5].

If LHCII phosphorylation was enhanced in SRB1, the conformation of the grana would be loosened, which could lead to a decrease in grana and an increase in unstacked stroma lamellar structures seen in the TEM images (Figure 3). Such a decrease in grana and increase in stroma lamella is also found under Fe-deficient EHM1 [18], suggesting that SRB1 can efficiently induce NPQ (Figure 1B) by retaining more mobile LHCII both under Fe-sufficient and -deficient conditions. Thus, we need to determine whether LHCII phosphorylation is more pronounced in SRB1 than in EHM1.

Another way to induce P700 oxidation is to promote electron transfer on the PSI acceptor side. The Calvin–Benson cycle, photorespiration, cyclic electron transfer flow (CEF), and the Water–Water cycle associated with ROS scavenging are essential for this in higher plants [19,24,25,27]. Our study did not investigate whether there are differences in these downstream PSI functions among cultivars. However, since ferredoxin (Fd) is depleted downstream of PSI in an Fe-deficient environment (Figure 2A), it is questionable whether Fd-mediated photorespiration and CEF are strongly induced in SRB1. The ROS scavenging pathways and the robustness and repair kinetics of PSI in SRB1 also warrant further analysis.

Despite MSS increasing the Y(NPQ) and Y(ND) under Fe deficiency similarity to SRB1, this cultivar showed the lowest Fe deficiency tolerance among barley varieties [6] and had a high Y(NA) value (Figure 1F), suggesting pronounced PSI photoinhibition during Fe deficiency. The reason PSI photoinhibition is high even though MSS could induce P700 oxidation is possibly due to its low Fe-usage efficiency within chloroplasts; MSS has the lowest photosynthetic Fe-use efficiency [6]. In fact, even under Fe-sufficient conditions, the Pm value (Figure 1D) and the accumulation of the Fe-binding protein PsaC was remarkably low in MSS (Figure 2A) even though its leaf-Fe content was higher than that of other cultivars (Appendix A). These results suggest that much Fe in MSS is not used to build PSI because the 4Fe-4S clusters are essential for de novo PSI assembly, but may be used for assembling other Fe-containing proteins, deposited in the tissue, or in a chemical form that cannot be recycled. Thus, the low efficiency of Fe supply to PSI in MSS would perturb the PSI maintenance even if the P700 oxidative system worked under Fe-deficient conditions.

Interestingly, SRB1 greatly decreased D1 and D2 accumulations than did EHM1 under Fe-deficient conditions, although the function of PSII of SRB1 was comparable to that of other cultivars (Figure 1). We calculated the ratio of signal intensity between Fe-deficient and control samples (Figure 2B, Figure 5E, and Appendix A). The loading amount in Figure 2B was normalized by protein amount; Figure 5E by proportions of the fractions, and Appendix A by chlorophyll amount. The results from the Western blot analysis showed that SRB1 rather decreased the accumulation of PSII core proteins. SRB1 could adopt a strategy of reserving small but functional amounts of reaction centers in the case of PSII and organizing ‘economical’ photosystems.

### 3.2. Sarab1 May Have a Characteristic Region of Thylakoid Membrane Supporting Smooth Electron Transfer under Fe-Deficient Conditions

The recovery rates of total thylakoid membranes based on chlorophyll from SRB1 leaves tended to be lower than that from EHM1 leaves regardless of Fe nutrition status, with a significant result observed in the case of Fe-deficient leaves (Appendix A, *p* = 0.16). Additionally, the rate of L-Thy to H-Thy of SRB1 was higher than that of EHM1 (Figure 5A,B). These findings do not contradict the observations that SRB1 leaves had more stromal thylakoid membranes, which may be fragile during the extraction procedure compared with EHM1 leaves using TEM (Figure 3). The structures and numbers of grana stack and stromal thylakoid membranes change in response to short- or long-term changes in light conditions, and these structural dynamics contribute to recovery from photoinhibition [28]. Nozue et al. reported that isolated grana, which were connected to a few stroma lamellae, exhibited a slower PSII repair than did stroma–grana structures [23]. Since the ability of photosystems to convert light energy into chemical energy decrease under Fe deficiency, Fe deficiency stress may result in a similar response to that of excess light conditions for chloroplasts, even under growth light conditions. Therefore, the higher rate of stromal thylakoid membranes in SRB1 chloroplasts may support the maintenance of photochemical reactions under Fe-deficient conditions. Functional differences between stromal membranes directly connected to granal membranes and stroma lamellae not directly connected to grana stacks were assumed [22]. Since we prepared segments for TEM using chemical fixation in this work, it was difficult to obtain a clear image of the whole chloroplast and to estimate a rate of two distinct stromal thylakoid regions using sufficient replicates. Quantitative comparison of the rate of grana stack, stromal thylakoid connected to grana, and not connected stroma lamellae among barley species adopting high-pressure freezing-freeze substituted fixation [29] may identify the advantageous structures of the thylakoid membrane under Fe-deficient conditions in the future.

The composition of H-Thy, which may correspond to well-characterized thylakoid membranes from SRB1 was highly similar to that from EHM1 based on the results from SDG centrifugation (Figure 4). Overall, the composition of L-Thy was similar to that of H-Thy, even though fr.7 corresponding to the PSI-LHCI fraction of L-Thy was reduced in EHM1 (Figure 4). Components of L-Thy in this work demonstrated that supernatants, which have not been previously analyzed, may contain functional thylakoid membranes. However, the dispositions of Fe and proteins to H-Thy and L-Thy were different between two Fe deficiency-tolerant cultivars. Almost the same amount of Fe remained in H- and L-Thy from SRB1 and EHM1 when barley plants were suffering from Fe deficiency (Figure 5C,D orange and orange hatched bars), while the accumulation of PSI reaction-center proteins on SRB1 thylakoid membranes was more drastically decreased by Fe deficiency than on EHM1 thylakoid membranes (Figure 2D and Figure 5E). Moreover, the rate of the distribution of Fe to L-thy was increased in SRB1 when compared to that in EHM1 (Figure 5C,D orange hatched bars). These data suggest that SRB1 thylakoid membranes contribute relatively larger amounts of Fe to the amounts of reaction-center proteins when compared to EHM1. However, the results from SDG centrifugation did not show the larger proportions of Fe in the fractions of PSI reaction-center proteins from SRB1 compared with those of EHM1 (Figure 4). Meanwhile, larger proportions of Fe were observed in the LHCII fractions of SRB1 than in those of EHM1 when grown under Fe-sufficient conditions (Figure 4 and Appendix A). We identified limited molecular species from each fraction using Western blot analysis in this work; thus, chemical species of Fe in the LHCII fraction are unknown. CBB staining of SDS-PAGE gels (Appendix A) showed differences in the protein composition of L-Thy between SRB1 and EHM1. The changes in thylakoid membrane structure discussed in 3.1, which also affect the arrangement and complex structures of PSII and PSI within the thylakoid membrane, may well explain the differences between SRB1 and EHM1 seen in fractionated thylakoid membrane samples. Thus, analyses of both proteins and Fe in intact protein complexes obtained from native PAGE are necessary in the future.

The results from SDG centrifugation suggest that the abundance of Lhcb1 in LHCII could be higher in SRB1 than in EHM1 (Figure 4 and Appendix A). This feature of SRB1 could be linked to a higher ratio of stromal/granal thylakoid (Figure 3). Furthermore, the rate of the distribution of Lhcb1 to L-Thy was largely increased by Fe deficiency both in SRB1 and EHM1 (Figure 5E). This data may correspond to the migration of Lhcb1 [18] and NPQ induction [6] under Fe-deficient conditions. Those Lhcb1s with high mobility in the unstacked thylakoid membranes could decrease the electron transfer from PSII in SRB1, probably forming some efficient energy quenchers around PSI, as we have suggested previously [5,17]. Indeed, the highest induction of NPQ was found in Fe-deficient SRB1 among barley cultivars (Figure 1B). Although NPQ is most often related to PSII photoprotection, NPQ also protects PSI, directly or indirectly through quenching part of the LHCII antenna pool functionally associated with PSI [30]. It is reasonable to assume that the Lhcb1-mediated NPQ induction in L-Thy of SRB1 resulted in higher 1-qL and Y(I)/Y(II) ratio (Appendix A) than in other cultivars, as well as in the strong induction of Y(ND) (Figure 1E) by Fe deficiency, leading to higher P700 oxidation levels in the Fe-deficient SRB1. Based on our data, we conclude that high P700 oxidation of Fe-deficient SRB1 could avoid the photoinhibition of PSI, allowing SRB1 to maintain PSI function even at low PSI content under prolonged Fe-deficient conditions.

Overall, L-Thy from Fe-deficient SRB1 leaves seems to have relatively high amounts of Fe, LHCII, PSII, and PSI proteins. These features are consistent with larger proportions of Fe in LHCII fractions from SDG fractionation. Details of associated proteins with the LHCII complex of SRB1 should be elucidated in the future. Based on the distribution of Fe and proteins on the thylakoid membrane described above, it is possible that SRB1 stocks some low-molecular-weight Fe polypeptide complexes fractionated to sucrose density with LHCII in L-thy, even when grown under Fe-sufficient conditions. Although many Fe proteins exist on thylakoid membranes, one of the candidate Fe proteins is PGR5, which belongs to the ferritin-like protein superfamily with Fe as a cofactor. Its primary role is CEF from PSI to the PQ pool, but its relevance to Fe partitioning on thylakoid membranes has also been discussed recently [31]. The speculation that such Fe-containing polypeptides might be important Fe reservoirs during Fe deficiency in low-density thylakoid membranes, such as stromal lamellae where PSI is localized, would be worth testing in the context of these recent discussions.

## 4. Materials and Methods

### 4.1. Plant Materials

Barley *Hordeum vulgare* L. cultivars and growth conditions were adopted for this study according to Saito et al. [6]. ‘Ehime Hadaka 1’ (EHM1), ‘Ethiopia 2’ (ETH2), ‘Musashinomugi’ (MSS), ‘Sarab 1’ (SRB1) were kindly provided by Professor Kazuhiro Satoh (Barley Germplasm Center, Okayama University, Japan). Seedlings were grown hydroponically in a growth chamber at 24/20 °C. The growth light intensity was set at 150–200 μmol photons m^−2^ s^−1^ under 14/10 h light/dark cycles. The control nutrient solution had 30 µM Fe-EDTA, and low Fe nutrient solutions were supplemented with 0.3–3 µM depending on the demand for Fe of each cultivar since a sufficient amount of thylakoid membrane cannot be obtained from severe chlorosis leaves. We used chlorotic leaves with a SPAD value (index of total chlorophyll content in a leaf area) of 15–20 as Fe-deficient leaves. Intact plants, fresh leaves, or leaves immediately frozen and stored at −80 °C were used for further experiments.

### 4.2. Measurement of Chlorophyll Fluorescence and P700 Redox State

Chlorophyll fluorescence and P700 redox state were measured simultaneously using a DUAL-PAM-100 (Heinz Walz GmbH, Effeltrich, Germany) in the young fully expanded leaves of 16 to 20-day-old plants as previously described [6]. The minimal fluorescence in the dark-adapted state (Fo) was recorded after the illumination of a weak measuring light (620 nm from the Dual-DR measurement heads) at a photon flux density of 5 μmol photon m^−2^ s^−1^. A saturating pulse (SP) light (300 ms, 14,000 μmol photon m^−2^ s^−1^) was applied to determine the maximal fluorescence in the dark-adapted state (Fm). The actinic light intensity increased in a stepwise manner (0, 6, 14, 32, 90, 168, 210, 326, 497, 755, 1174 μmol photons m^−2^ s^−1^), and the actinic light condition at 210 μmol photons m^−2^ s^−1^ was used for the main analysis as growth light conditions. The maximal and minimal fluorescence in the light-adapted state (Fm’ and Fo’) and steady-state chlorophyll fluorescence (Fs) were recorded during the exposure to the actinic light illumination. Prior to measuring Fm′ and Fo′, a saturating pulse light and far-red light (720 nm) were applied, respectively. The maximal quantum yield of PSII and NPQ were calculated as Fv/Fm and (Fm − Fm′)/Fm′, respectively. The index for the reduction in the primary PQ electron acceptor (QA) (1−qL) was calculated as 1 − (Fm′ − Fs)/(Fm′ − Fo’) × (Fo′/Fs). Y(II), Y(NO), and Y(NPQ) were calculated as (Fm′ − Fs)/Fm’, 1/[NPQ + 1 + qL(Fm/Fo − 1)], and 1 − Y(II) − 1/[NPQ + 1 + qL(Fm/Fo − 1)], respectively.

Simultaneous with the chlorophyll fluorescence measurements, the redox change of P700 was assessed by monitoring the changes in the absorbance of transmission light at 830 and 875 nm, according to Klughammer and Schreiber [32]. The maximal P700 signal (Pm) was determined by applying a saturated pulse light in the presence of far-red light (720 nm), while that of the oxidized P700 during actinic light illumination (Pm’) was determined by the saturated pulse-light application. The P700 signal during actinic light illumination (P) was recorded just prior to the saturated pulse light application. Y(I), Y(NA), and Y(ND) were calculated as (Pm′ − P)/Pm, (Pm − Pm′)/Pm, and P/Pm, respectively.

To determine the functional PSI from the value of Pm, the absolute Pm value, which is affected by both PSI content and PSI function in leaves, was divided by the value of the relative amounts of PSI core proteins, PsaA, PsaB, and PsaC, respectively. The obtained Pm/PsaA, Pm/PsaB, and Pm/PsaC values are the ‘Normalized Pm’, which is correlated to absolute functional PSI. Finally, the ratio (‘Normalized Pm’ of Fe-deficient leaves)/(‘Normalized Pm’ of Fe-sufficient leaves) was calculated as the functional PSI ratio under Fe-deficient conditions shown in Figure 2D and Appendix A.

### 4.3. Transmission Electron Microscopy (TEM) and Measurement of Granal and Stromal Thylakoids

We adopted and modified the methods described in Saito et al. [18]. Barley leaves were cut with a razor to squares of lengths less than 2 mm, and fixing solution (2% paraformaldehyde, 2% glutaraldehyde in 0.05 M cacodylate buffer pH 7.4 at 4 °C) was rapidly added to the leaves for absorption in a sealed syringe by pulling the piston to reduce pressure. After incubation in the fixing solution at 4 °C overnight, the samples were washed three times with 0.05 M cacodylate buffer for 30 min each. Samples were postfixed with 2% osmium tetroxide (OsO_4_) in 0.05 M cacodylate buffer at 4 °C for 3 h. After dehydration with ethanol series and infiltration with propylene oxide, samples were embedded in resin Quetol-651 (Nisshin EM Co., Tokyo, Japan) and polymerized. The polymerized resins were ultra-thin sectioned to 80 nm using an ultramicrotome Ultracut UCT (Leica, Vienna, Austria), and sections were stained with 2% uranyl acetate followed by secondary staining with a lead stain solution (Sigma-Aldrich Co., St. Louis, MO, USA). The grids were observed using a JEM-1400 plus transmission electron microscope (JEOL Ltd., Tokyo, Japan) at an acceleration voltage of 80 or 100 kV. Images were taken with a CCD camera VELETA (Olympus Soft Imaging Solutions GmbH, Münster, Germany) or EM-14830RUBY2 (JEOL Ltd.).

Granal and stromal thylakoid membranes were estimated using NIH ImageJ 1.53t software (https://imagej.nih.gov/ij/, accessed on 24 August 2022). We sampled four independent chloroplast images for each cultivar and selected all grana stacks with lamellar membranes clearly recognized in each image. Each electron-dense layer in grana stacks was traced by two lines as granal thylakoid membranes. Thylakoid membranes connected to selected grana stacks were traced as stromal thylakoid membranes. Summations of line lengths belonging to grana or stroma were calculated for each image, then stroma/grana ratios were presented.

### 4.4. Thylakoid Membrane Extraction

Fresh or frozen leaf pieces were homogenized with an isotonic solution and filtered crude thylakoid samples were corrected as pellets by centrifugation at 2500× *g* at 4 °C for 10 min, as described by Saito et al. [6]. We designated green pellets suspended in the buffer obtained by 2500× *g* centrifugation as “heavy/high-density thylakoids (H-Thy)”. The supernatants after centrifugation containing considerable amounts of chlorophyll were subjected to further centrifugation at 10,000× *g* at 4 °C for 10 min, and these green pellets were designated as “light/low-density thylakoids (L-Thy)”. Precipitates were resuspended in HM buffer [6] and stored at −80 °C as H-Thy and L-Thy fractions.

### 4.5. Sucrose Density Gradient (SDG) Centrifugation

Discontinuous SDG tubes were prepared by sequential layering of 1.3 M, 1.0 M, 0.7 M, 0.4 M, and 0.1 M sucrose (from bottom to top) with 0.05% (*w*/*v*) of β-DM. Thylakoid membrane samples as 0.4 mg/mL of chlorophyll were solubilized with β-DM (4.8 mg β-DM for 0.1 mg chlorophyll) at 4 °C in the dark for 2 min. The solubilized sample was loaded on the top of the SDG tube and centrifuged at 280,000× g for 20 h using a P40ST rotor (Eppendorf Himac Technologies Co., Ltd., Ibaraki, Japan) at 4 °C.

### 4.6. Measurement of Chlorophyll, Proteins, and Fe

Chlorophyll in thylakoid membranes was quantified after resuspending aliquots of samples in 80% (*v*/*v*) acetone [33]. The amounts of proteins in each sample were quantified using the BCA (bicinchoninic acid) method with Protein Assay Standard I (Bio-Rad Laboratories, Inc, Hercules, CA, USA). Dried leaves were digested in concentrated HNO_3_ at 150 °C and dissolved in 1% (*v*/*v*) HNO_3_. Thylakoid fractions were digested using extremely clean reagents, instruments, and atmosphere according to Saito et al. [6], due to trace amounts of Fe in thylakoid membranes. The Fe concentration was measured using an atomic absorption spectrophotometer (AA-6300, Shimadzu, Tokyo, Japan) coupled with a graphite furnace atomizer (GFA-EX7i, Shimadzu, Tokyo, Japan).

### 4.7. Western Blot Analysis

Proteins of the thylakoid membranes were solubilized in protein-solubilizing buffer and subjected to SDS–PAGE. Immunoblot analysis was performed as described previously [17]. Anti-PsbA/D1 (AS05-084), anti-PsbB/D2 (AS06-146), anti-PsbE/cyt *b*_559_ (AS06-112), anti-Lhcb2 (AS01-003), anti-PsaA (AS05-084), anti-PsaB (AS10-695), anti-PsaC (AS10-939), and anti-FDX1/ferredoxin (AS06-121) antibodies were obtained from Agrisera (Vännäs, Sweden). Anti-HvLhcb1 antibody was raised against the peptide derived from HvLhcb1 by Eurofins Genomics (Tokyo, Japan). Signal intensities were quantified using NIH ImageJ 1.53t software with plug-in ‘BandPeakQuantification’ [34].

## 5. Conclusions

We found that the Fe deficiency-tolerant barley cultivar SRB1 has a characteristic region on the thylakoid membrane that contains a considerable amount of Fe and specific composition of photosystem proteins when compared to the corresponding region in EHM1. Such a region on the thylakoid membrane may contribute to effective electron transfer utilizing only a small amount of Fe and reaction-center proteins under Fe-deficient conditions. The survey of the composition of photosystem proteins among various barley cultivars and QTL analysis are ongoing. The advantages of such an organization of the thylakoid membrane of SRB1 under other stress conditions will be of interest to understand the acclimation of photosynthetic apparatus with high plasticity.

## Figures and Tables

**Figure 1 plants-12-02111-f001:**
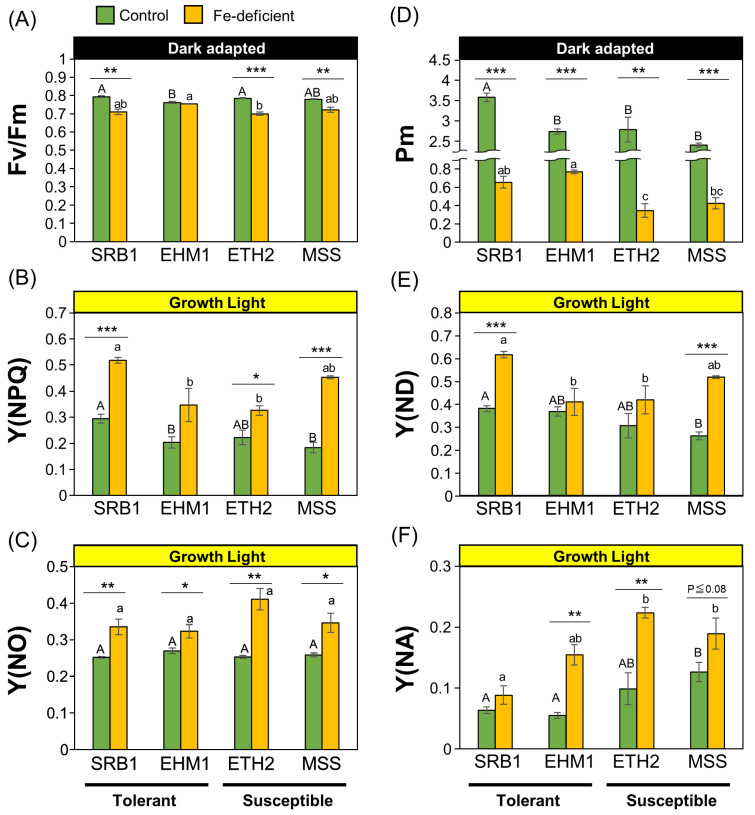
Comparison of the functionality of PSII and PSI in four barley cultivars with varied Fe deficiency tolerance levels: PSII maximum quantum yield, Fv/Fm (**A**); the quantum yield of light-induced non-photochemical fluorescence quenching induced for photoprotection of PSII, Y(NPQ) (**B**); the quantum yield of non-light-induced non-photochemical fluorescence quenching related to PSII photoinhibition; Y(NO) (**C**); the maximal P700 signal, Pm (**D**); PSI donor-side electron transfer limitation, Y(ND) (**E**); PSI acceptor-side electron transfer limitation, Y(NA) (**F**). Data are represented as the means ± SE of three to four independent measurements. * *p* < 0.05, ** *p* < 0.01, and *** *p* < 0.001, indicate significant differences between +Fe and −Fe treatments (according to Student’s *t*-test). Different letters are shown on individual columns when *p* is <0.05 among the four barley cultivars based on Tukey multiple testing.

**Figure 2 plants-12-02111-f002:**
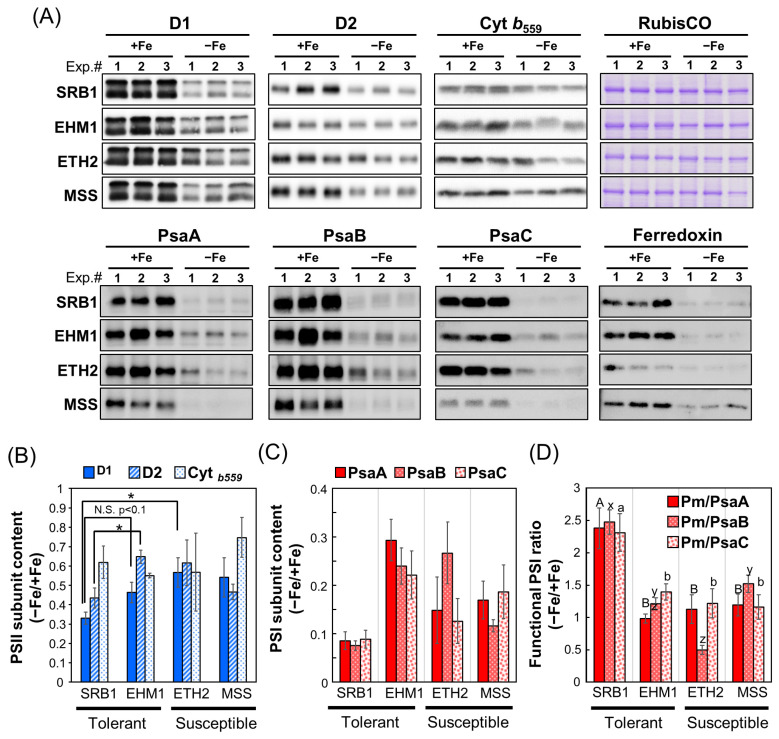
Western blot analysis normalized on a per total leaf protein to compare whole thylakoid proteins and functional PSI levels in four barley cultivars with different Fe deficiency tolerance: (**A**) Immunoblot analysis of PSII reaction center proteins (D1, D2, and cyt *b*_559_ [PsbE]) and PSI reaction center proteins (PsaA, PsaB, and PsaC), Ferredoxin, and CBB-stained RubisCO large subunits. Whole proteins extracted from leaves were separated by SDS-PAGE (1 μg protein/lane for D1, 5 μg protein/lane for the other proteins) and detected with specific antibodies for each protein; (**B**,**C**) Immunoblots detected with specific antibodies against each PSII subunit (D1, D2, and cyt *b_559_*) (**B**) or PSI subunit (PsaA, PsaB, and PsaC) (**C**) in panel A was quantified by Image J software and calculated as relative values for the Fe-sufficient condition (Fe-sufficient condition = 1); (**D**) The retention rate of functional PSI under the Fe-deficient condition is expressed as the relative value of Pm per PSI subunit content under Fe-deficient conditions (value of Figure 2C) to that under Fe-sufficient conditions. Data are presented as means ± SE of three independent leaves, with different letters shown on individual columns when *p* < 0.05 among four barley cultivars based on Tukey multiple testing. * *p* < 0.05.

**Figure 3 plants-12-02111-f003:**
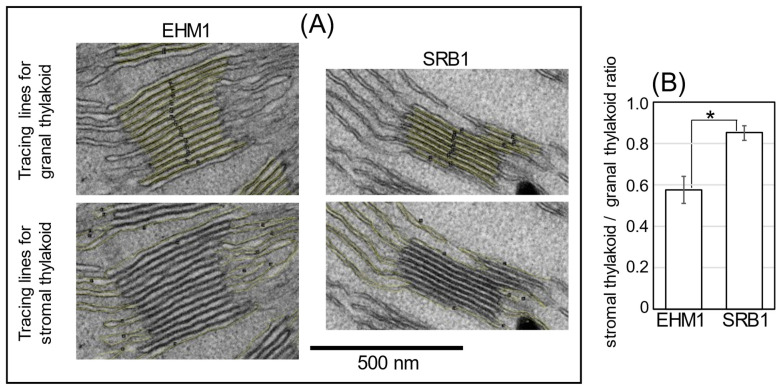
The ratio of stromal and granal thylakoids in Fe-sufficient chloroplasts. TEM images of Fe-sufficient chloroplasts were obtained from SRB1 and EHM1. All original images are shown in Appendix A. We sampled four independent chloroplasts images per cultivar, and selected grana stacks with stroma lamellar membranes clearly recognized in each image. Then, granal thylakoid and thylakoid membranes in stroma connected to selected grana were measured by line length using ImageJ. Typical magnified images of yellow lines tracing the membrane of grana and stroma are shown in (**A**). All tracing lines on membranes are shown in Appendix A. Summations of line lengths to grana or stroma were calculated for each image, then averages of the stromal thylakoid/granal thylakoid ratio, as shown in (**B**). Values represent the mean ± SE of the four images. * *p* < 0.05 indicates significant differences (according to Student’s *t*-test) between two cultivars.

**Figure 4 plants-12-02111-f004:**
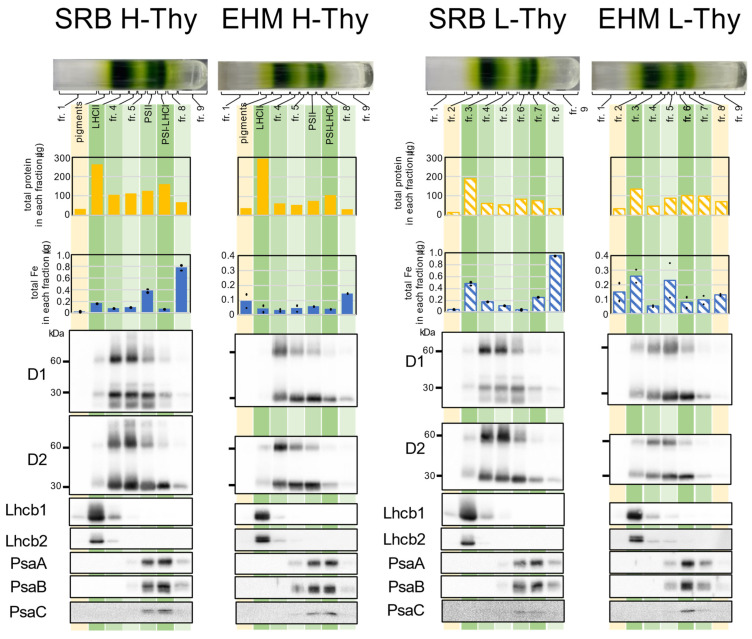
Analyses of fractions obtained from SDG using the thylakoid membranes derived from SRB1 and EHM1. Thylakoid membrane samples from 0.4 mg/mL of chlorophyll were solubilized with β-DM (4.8 mg β-DM for 0.1 mg chlorophyll) and loaded on the top of the SDG tube. Total amounts of proteins or Fe in each fraction are presented in the bar graphs. The amount of Fe was determined by two test solutions prepared from one fraction and measurement was conducted twice for one test solution. A dot shows the average of one test solution and a bar shows the average of two test solutions. A total of 1/1000 of each fraction was loaded for Western blot analysis, except for anti-PsaC. For anti-PsaC, 1/500 of each fraction was loaded.

**Figure 5 plants-12-02111-f005:**
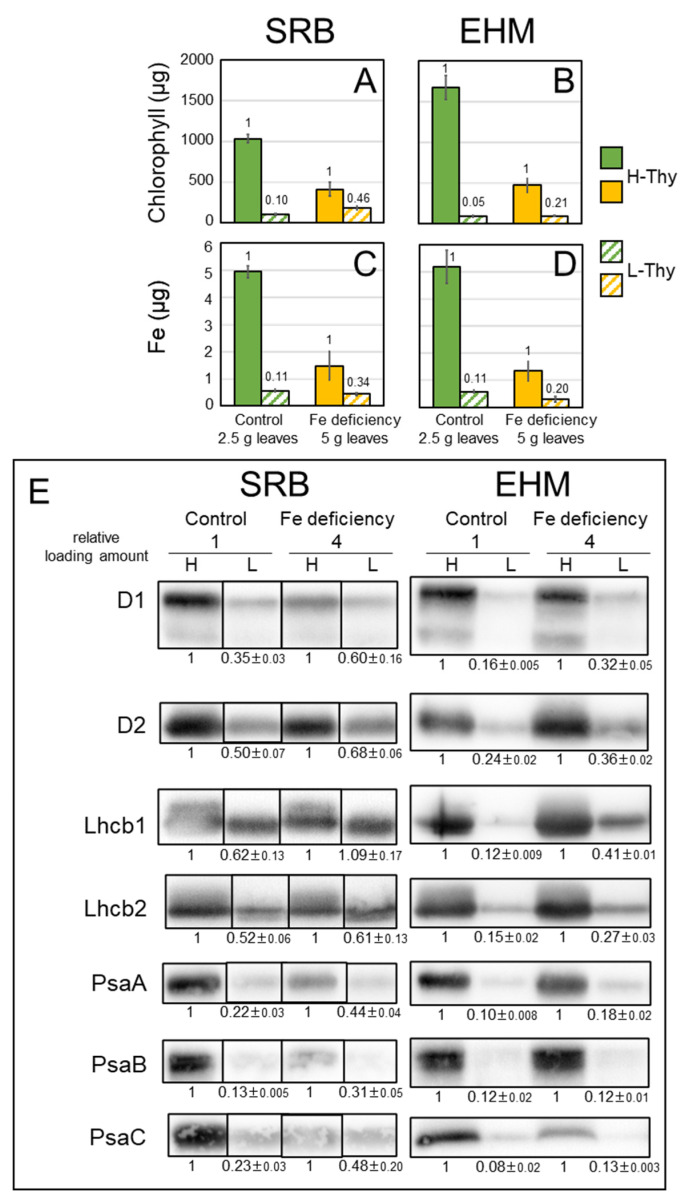
Chlorophyll, Fe, and proteins present on high- and low-density thylakoids from SRB1 and EHM1. (**A**,**C**) SRB1 and (**B**,**D**) EHM1. (**A**,**B**) Chlorophyll content and (**C**,**D**) Fe contents derived from 2.5 g or 5 g of control or Fe-deficient leaves, respectively. Solid bars and hatched bars represent H-Thy and L-Thy, respectively. Values represent the mean ± SE of three independent fractions. Small counts on bars indicate the ratios of L-Thy to H-Thy. (**E**) Western blot analysis. Loading amounts: 1/2000 of the fraction for the control sample and 1/500 of the fraction for the Fe-deficient sample, except anti-PsaC. For anti-PsaC, 1/1000 or 1/250 fractions were loaded for the control sample or Fe-deficient sample, respectively. Higher amounts of Fe-deficient materials than control materials were used to obtain a clear signal. Lanes of images from SRB1 were rearranged to match the order of those from EHM1. Corresponding CBB stain images are shown in Appendix A, and original blot images of three replicates are shown in Appendix A. Relative signal intensities of L-Thy to corresponding H-Thy were calculated and average and SE (*n* = 3) were shown.

## Data Availability

Not applicable.

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
