# Peer review of "Barley Cultivar Sarab 1 Has a Characteristic Region on the Thylakoid Membrane That Protects Photosystem I under Iron-Deficient Conditions"

_plants, 2023, doi:10.3390/plants12112111_

Round 1
Reviewer 1 Report
The manuscript ‘Barley cultivar Sarab 1 has a characteristic region of thylakoid membrane and protect photosystem I under the iron-deficient condition’ by Saito et al claims that SRBi has non-appressed thylakoid membranes and large portion of functional PSI, the reasons for Fe-deficient ability of SRB1. This is a continuous study after Saito et al 2021 (ref 6).
However, some data and explanation does not support each other. Lines 137-139, authors claimed ‘similar … between fig 1E and Fi B? are they?
From the data in Fig S2A, authors claimed ‘YI/YII of SRBI is special and efficient to PSI electron transport, were authors mean electron flow from PSII to PSI?
Line 158, should read as Fig S3? The statement is not matching with cited Fig S8.
The figures also showed MSS’ s decreased PSII under iron-stressed conditions, The statement in lines 156 – 182, Authors indicated the data selectively, which may be mislead readers.
Data in Fig 2 and Fig S3, are different (seems the same experiments), very confusion. The Fig and some explanation/statement are different. E.g Fig 2, Western, PsaA/A/C showed similar response under -Fe condition between SRB1 and MSS.
Reviewer cannot judge Fig 3 and its related results/conclusion because no support data/images for it. The figure 3 legend did not give enough information to understand the figure. What is Panel A and Panel B (from which plant, how many images were recorded for Fig 3C?
As author claimed, the ‘distribution and portion of Fe ‘(Fig 4 and S5) not reproducible, not sure how many biological repeats and what error bars represents? Very difficult to digest the data and understand what authors claims. If not reproducible data, not sure, how to access the results (line 281-295). Authours should consider to eliminate the errors with a few repeatable data-set.
Fig 5 should include MSS as control.
NPQ and LHCII are mainly associated with the function of PSII, However, authors in this paper wish to claim that ‘Large Portion functional PSI’ is the reason for SRB1 Fe-tolerance. Very confusion. Therefore, author should explain the relationship between NPQ/LHCII data and functional PSI. (how to define ‘functional PSI’).
Author should consider to re-edit Fig 1 to focus on "PSI".
Author Response
Please refer attached PDF file.

Reviewer 2 Report
This study compared activities of photosynthetic electron transfer, organization of thylakoid membranes, and distribution of Fe and proteins on thylakoid membranes between two barley cultivars. In general, the research was well-performed. It may be published pending some moderate revisions.
1) All abbreviations should be introduced at the first time of appearance, such as the cultivar names "ETH2 and MSS".
2) The language is poor and sometimes confusing. For example, in Lines 18-20, "using transmission electron microscopy and fractionation of thylakoid membranes by specific gravity revealed that SRB1 has non appressed thylakoid membranes containing a larger amount of Fe and LHCII proteins when compared to EHM1." What gravity? centrifugal force? You mean that EHM1 does not have non-appressed thylakoid membranes?
In Lines 22-23, "EHM1 may insert Fe cofactors to PSI exploiting surplus reaction center proteins under the Fe-deficient condition." What does the word "insert" mean? How to insert?
3) SRB1 is sensitive to Fe-deficient conditions, however it contains a higher amount of Fe and LHCII proteins (from Figure 5, I cannot draw this conclusion). How do you explain the discrepancy?
4) In Figure 3, the scale bar is missing. By the way, I cannot see the difference between the two cultivars.
5) In Figure 5, Only "H" sample in the control treatment may be normalized into 1 (100%), but not the "H" sample under the Fe-deficiency. Then multiple comparisons should be performed.
See the comment #2.
Author Response
Please refer attached PDF file.

Round 2
Reviewer 1 Report
Authors made significant improvement in the revision. a few suggestions and comments here:
Page 3 Lines 112 – 120. How authors to define the ‘significance’ or only ‘slight reduction’ or ‘increase significantly’ Fig 1B, from ~02 -0.3x defined as increased significantly, but Fig 1C from ~0.25 to 0.4 defined as ‘no significant difference’. Need to define the statement with ‘good’ standard. Here is not the results from the data in the figure, instead, authors used their ‘statement’ to guide readers.
May need more information for ‘Thus, wee confirmed that PSII photoinhibition is not …’ (Page 3, line 122-123).
Fig 1B, MSS showed the highest increased Y(NPQ) although it has slightly lower than SRB1 under iron deficient condition? Is this observation significant than authors’ statement ‘SRB1 … highest Y(NPQ)..’.
According to data in Sup Fig S1B, under iron-deficient condition, SRB1 and MSS showed similar iron content per leaf unit, considering PSI contents, relative high iron content /PSI unit in MSS, less stressed?
The reason for comparing the ratio of stromal and granal thylakoid between SRB1 and EHM1, instead of SRB1 vs MSS or EHM1 vs MSS?
If PSII most likely were localised in the granal region, with less PSII in EHM1 (supported by fig 2 and S3), the EM data seems disagree with the protein analysis?
Author Response
Please refer the attached PDF file.
